# Recent Treatment Strategies and Molecular Pathways in Resistance Mechanisms of Antiangiogenic Therapies in Glioblastoma

**DOI:** 10.3390/cancers16172975

**Published:** 2024-08-27

**Authors:** Md Ataur Rahman, Meser M. Ali

**Affiliations:** Department of Oncology, Karmanos Cancer Institute, Wayne State University, Detroit, MI 48201, USA

**Keywords:** glioblastoma (GBM), angiogenesis, resistance, immunotherapy, antiangiogenic treatments

## Abstract

**Simple Summary:**

Glioblastoma (GBM) appears to be a challenging malignancy to completely eradicate, with few significant improvements in treatment. This review explores the mechanisms by which GBM tumors acquire resistance to antiangiogenic therapy, which are designed to inhibit the growth of blood vessels in tumors. This study addresses numerous mechanisms of resistance, including redundant pathways, heightened invasion, hypoxia, and immune modulation. This article additionally highlights potential strategies to overcome resistance, including combination therapies, personalized medicine, immunotherapy, and drug delivery using nanoparticles, with the goal of enhancing treatment results for patients with GBM.

**Abstract:**

Malignant gliomas present great difficulties in treatment, with little change over the past 30 years in the median survival time of 15 months. Current treatment options include surgery, radiotherapy (RT), and chemotherapy. New therapies aimed at suppressing the formation of new vasculature (antiangiogenic treatments) or destroying formed tumor vasculature (vascular disrupting agents) show promise. This study summarizes the existing knowledge regarding the processes by which glioblastoma (GBM) tumors acquire resistance to antiangiogenic treatments. The discussion encompasses the activation of redundant proangiogenic pathways, heightened tumor cell invasion and metastasis, resistance induced by hypoxia, creation of vascular mimicry channels, and regulation of the tumor immune microenvironment. Subsequently, we explore potential strategies to overcome this resistance, such as combining antiangiogenic therapies with other treatment methods, personalizing treatments for each patient, focusing on new therapeutic targets, incorporating immunotherapy, and utilizing drug delivery systems based on nanoparticles. Additionally, we would like to discuss the limitations of existing methods and potential future directions to enhance the beneficial effects of antiangiogenic treatments for patients with GBM. Therefore, this review aims to enhance the research outcome for GBM and provide a more promising opportunity by thoroughly exploring the mechanisms of resistance and investigating novel therapeutic strategies.

## 1. Introduction

Glioblastoma (GBM) is the predominant and highly malignant primary brain tumor in adults, known for its rapid proliferation, invasive nature, and unfavorable prognosis [1,2]. Despite the progress made in surgical methods, radiotherapy, and chemotherapy, the average lifespan of individuals with GBM remains disappointingly short, usually approximately 15 months after being diagnosed [3]. Many promising therapies target microvasculature in this highly vascular tumor. These fall into two main categories: antiangiogenic (AA) agents [4,5,6,7,8,9], which inhibit the formation of new vasculature, and vascular disrupting agents (VDAs), which selectively destroy the tumor vasculature. Antiangiogenic therapies often target the proangiogenic compound vascular endothelial growth factor (VEGF), which is widely expressed in gliomas [10,11], or VEGF receptors (VEGFRs). Since VEGF is known to increase vascular permeability in mature vessels and promote the proliferation of new leaky microvasculature, Vredenburgh [4] et al. have demonstrated in a phase II trial that bevacizumab, a VEGF-A inhibitor, extends the progression-free survival in patients with recurrent malignant gliomas by a factor of two or more. It was noted therein that two-thirds of the patients showed a partial response, which was judged by a greater than 50% decrease in the area of contrast enhancement on T_1_-weighted post-contrast images [12]. The reliance on angiogenesis has resulted in the emergence and application of antiangiogenic treatments that target the suppression of new blood vessel production [13]. This effectively deprives the tumor of the necessary nutrition and oxygen it needs to grow and survive. Antiangiogenic treatments, including those that focus on inhibiting VEGF and its receptors, demonstrated initial potential in preclinical models and early clinical studies [14]. Bevacizumab, a monoclonal antibody that targets VEGF, has shown notable enhancements in the length of time before disease progression and alleviation from symptoms in individuals with GBM [15]. Nevertheless, the excitement around these treatments has been dampened by the recognition that their advantages are frequently short-lived and do not substantially prolong overall lifespan. Tumors often acquire resistance to antiangiogenic therapies, which reduces their long-term effectiveness and presents a major obstacle in the clinical treatment of GBM [16]. The conventional approach usually entails the utilization of a blend of surgical intervention, radiation therapy, and chemotherapy employing temozolomide [17]. Nevertheless, these treatments generally focus on extending lifespan and enhancing the overall well-being rather than offering a complete remedy.

GBM tumors utilize redundant angiogenic pathways as a main means to resist antiangiogenic therapy [18]. Tumor cells could increase the production of other substances that promote the formation of blood vessels, such as fibroblast growth factor (FGF), platelet-derived growth factor (PDGF), and angiopoietins, when the VEGF signaling is blocked [19]. These alternate routes can offset the decrease in VEGF function, maintaining the formation of new blood vessels and the growth of tumors [20]. Furthermore, inhibiting VEGF can unexpectedly enhance the invasiveness and ability to spread to other parts of the body in GBM cells [21]. The heightened invasiveness that ensues enables tumor cells to spread into the adjacent brain tissue, resulting in recurrences that are resistant to treatment [22]. Hypoxia, which refers to a state of decreased oxygen levels in the tumor microenvironment, is an additional important component contributing to resistance against antiangiogenic treatments [23]. Antiangiogenic therapy may aggravate hypoxia by interrupting the tumor’s blood supply, leading to the activation of hypoxia-inducible factors (HIFs) [24]. These transcription factors control the activity of several genes that are involved in the processes of angiogenesis, metabolism, and survival [25]. As a result, it facilitates the tumor’s ability to adapt and develop resistance. In addition, hypoxia can induce the occurrence of vascular mimicry, which refers to the formation of vessel-like structures by tumor cells themselves, enabling blood flow without the involvement of endothelial cells [26].

Considering the resistance mechanisms, current treatment techniques are adapted to address the complex character of GBM. Researchers are currently investigating combination therapies that simultaneously target various angiogenic pathways. They are also studying therapies that combine antiangiogenic drugs with conventional chemotherapy, radiation, or immunotherapy [16]. Personalized medicine strategies, utilizing molecular profiling of cancers to customize treatments for individual patients, show potential in improving therapeutic results [27]. In addition, researchers are discovering additional therapeutic targets that go beyond the conventional pathways involved in angiogenesis. These findings present new opportunities for intervention. Investigations are also underway to explore the utilization of nanoparticles to enhance drug delivery and effectiveness, which is considered another pioneering strategy [28]. Although there has been notable progress in this area, there are still substantial obstacles to overcome in addressing the resistance to antiangiogenic treatment in glioblastoma. The diverse and flexible nature of the tumor, together with the intricate interactions within the tumor microenvironment, need a comprehensive and ever-changing strategy to treatment [29]. In this current review, we would like to consider understanding the complex processes of resistance and discussing combination therapy that can effectively target the robust characteristics of GBM.

## 2. Mechanisms of Resistance to Antiangiogenic Therapies in GBM

Glioblastoma (GBM) is a remarkably aggressive brain tumor characterized by its extensive network of blood vessels [30]. Antiangiogenic therapies seek to disrupt the blood supply by specifically targeting VEGF, a crucial element in the process of angiogenesis. Nevertheless, GBM tumors acquire resistance through many methods, as shown in Figure 1:

### 2.1. Redundant Angiogenic Pathways

Angiogenesis is a defining characteristic of GBM and continues to be a significant focus in its therapy, particularly for cases with recurring GBM, a crucial regulator and stimulator of angiogenesis. Hence, the development of antiangiogenic treatments (AATs) that specifically target VEGF or VEGF receptors (VEGFRs) was undertaken with the aim of effectively managing the growth of GBM [31]. The redundancy and intricacy of angiogenic pathways in GBM provide substantial obstacles to the effectiveness of antiangiogenic treatments [1]. Alternative proangiogenic factors can be activated by tumor cells when VEGF is inhibited, leading to the promotion of blood vessel formation through alternative signaling pathways [32]. The factors include FGF, PDGF, HGF, angiopoietins (Ang), and interleukins (ILs) [33]. These substances can promote the growth and movement of endothelial cells, leading to the formation of new blood vessels without relying on VEGF [34]. Anti-VEGF therapies may selectively focus on stages; however, GBM could trigger alternate pathways for other stages [35]. For instance, certain therapies specifically focus on inhibiting the binding of VEGF to its receptors. Tumors may increase the expression of alternative receptor signaling or employ non-receptor mediated methods to promote angiogenesis [36]. However, the presence of low oxygen levels, known as hypoxia, resulting from antiangiogenic therapy, can stimulate the development of alternative pathways for blood vessel formation in the TME [13]. Hypoxia triggers the activation of HIF-1α, a key controller that stimulates the production of many proangiogenic factors in addition to VEGF. This results in the formation of intricate signaling pathways that could circumvent the inhibition of VEGF [37]. Gaining an understanding of duplicated pathways is essential since it elucidates the reason why solely targeting VEGF frequently proves ineffective in managing GBM progression (Figure 2). Tumors utilize these pathways to adjust and generate fresh blood vessels, hence undermining the efficacy of antiangiogenic treatments [38]. Scientists are investigating methods to overcome this barrier. Current research is focused on exploring combination therapies that can simultaneously target various pathways involved in angiogenesis, and pharmaceuticals that specifically target the signaling molecules activated by these pathways [19]. Sunitinib is a small molecule multitarget receptor tyrosine kinase inhibitor that can block signaling through various receptors, including PDGFRs, VEGFRs, c-KIT, colony-stimulating factor-1 receptor, and fetal liver kinase 3-internal tandem duplication (FLT3-ITD) [31]. Furthermore, researchers are investigating methods to standardize and enhance the functionality of pre-existing blood arteries. Understanding these pathways is essential for developing more efficient therapeutic approaches that can overcome resistance and enhance patient outcomes. Subsequent investigations should prioritize the development of treatment strategies that combine therapies targeting several pathways involved in the formation of new blood vessels and the surrounding environment of the tumor, with the goal of achieving long-lasting suppression of tumor growth.

### 2.2. Increased Invasion and Metastasis

GBM is a highly aggressive brain tumor known for its resistance to many therapies, including antiangiogenic therapies. An important factor contributing to the resistance of these therapies is the heightened invasion and spread of the tumor cells [39]. Antiangiogenic therapies aim to hinder the blood supply of the tumor by blocking the activity of VEGF and other pathways involved in the formation of new blood vessels (angiogenesis) [13]. Although this technique initially decreases the blood vessel formation and growth of the tumor, it unintentionally encourages a more aggressive behavior of the tumor. Application of antiangiogenic therapy induces a hypoxic environment in the tumor by decreasing blood flow, which in turn stimulates various adaptive responses [13]. Under situations of low oxygen, HIFs become stable, resulting in the activation of genes that promote cell survival, invasion, and metastasis [40]. For example, HIF-1α enhances the production of matrix metalloproteinases (MMPs) and other enzymes that break down the extracellular matrix (ECM), enabling tumor cells to more efficiently penetrate the surrounding brain tissue [23].

Furthermore, the invasive characteristics of GBM cells are influenced by the process of hypoxia-induced epithelial–mesenchymal transition (EMT) [41]. During the process of EMT, tumor cells undergo a loss of their epithelial characteristics, including cell–cell adhesion, and acquire mesenchymal traits, which leads to an increase in their ability to move and invade surrounding tissues [42]. This metamorphosis is partially influenced by factors related to low oxygen levels (hypoxia) and signaling pathways involving growth factors, such as transforming growth factor-TGF-β and HGF [43]. In addition, the oxygen-deprived environment might stimulate the production of chemokines and their receptors, such as CXCL12 and CXCR4, which in turn promote the movement and infiltration of tumor cells [44]. Chemokine interactions facilitate the evasion of immune surveillance by tumor cells and the establishment of secondary sites of growth, hence promoting metastasis [45]. Ultimately, the hypoxia caused by antiangiogenic therapy can result in a TME that is both more aggressive and adaptable [23]. The hypoxic conditions exert selection pressure, leading to the creation of highly invasive subclones of tumor cells that can thrive despite the inhospitable environment [46]. The presence of this adaptive resistance mechanism emphasizes the difficulty in treating GBM and emphasizes the necessity for combined therapies that address both angiogenesis and the invasive characteristics of the tumor cells [46]. Scientists are currently studying methods to address this issue, such as merging antiangiogenic therapy with medications that focus on invasion pathways or utilizing targeted therapies that regulate the tumor vasculature without stimulating invasion.

### 2.3. Hypoxia-Induced Resistance

A major challenge in the management of GBM is the development of resistance to various medications due to hypoxia. GBM often exhibits hypoxia, which is characterized by low oxygen levels in the tumor microenvironment [47]. Tumor growth occurs because of the rapid multiplication of tumor cells surpassing their blood supply and the abnormal blood vessels that are typical of these tumors [48]. Hypoxia initiates a series of adaptive reactions that lead to the development of resistance to therapy [24]. The HIFs, specifically HIF-1α and HIF-2α, are the main individuals involved in this process [49]. Transcription factors are made more stable in low oxygen circumstances and trigger the activation of several genes that support the survival and adaptability of tumors [50]. Hypoxia increases resistance to antiangiogenic therapy by upregulating proangiogenic proteins, including VEGF, angiopoietins, and FGF [51]. Although antiangiogenic medications suppress VEGF signaling pathways, hypoxia can still promote angiogenesis through other mechanisms, which allows the tumor to maintain its blood supply and continue growing and surviving [51]. In addition, hypoxia stimulates a more assertive and infiltrative tumor phenotype. Hypoxia-induced activation of HIF leads to the upregulation of MMPs and other enzymes that breakdown the extracellular matrix, hence promoting the migration of tumor cells into the adjacent healthy brain tissue [52]. This intrusive tendency also complicates treatment, as invasive tumor cells are less reachable for both surgical removal and localized therapies. Hypoxia also affects the immune response in the tumor microenvironment [23]. It can stimulate the production of immunological checkpoint molecules and modify the behavior of immune cells, which leads to a suppressive environment that impairs the effectiveness of immunotherapies [53]. Furthermore, the metabolic reprogramming of tumor cells caused by hypoxia promotes their ability to survive in settings when nutrients are scarce, hence increasing their resistance to therapies that target metabolic pathways [54]. Overall, resistance to hypoxia in GBM is a complex interaction that includes heightened angiogenesis, elevated invasion, evasion of the immune system, and metabolic adaptability [55]. Gaining a comprehensive understanding of these pathways is of utmost importance for devising more efficient therapeutic approaches that can effectively address the difficulties presented by hypoxia in GBM [56]. To significantly improve the efficacy of antiangiogenic therapies and potentially enhance patient outcomes for GBM, it is necessary to overcome the resistance caused by hypoxia.

### 2.4. Vascular Mimicry

Vascular mimicry (VM) is the formation of blood vessel-like structures in extremely aggressive tumors, such as GBM, without the presence of endothelial cells [57]. Tumors can sustain their growth and spread to other parts of the body by bypassing standard mechanisms of blood vessel formation [58]. This poses considerable problems to therapies that aim to prevent the establishment of new blood vessels. GBM is characterized by tumor cells undergoing a transformation where they acquire characteristics like endothelial cells [59]. This transformation allows the tumor cells to create vascular channels [60]. Tumor cells in this phase display endothelium markers, including CD31, VE-cadherin, and Factor VIII [61]. Transcription factors like as Ets-1 and HIF-1α are essential in controlling this change in phenotype [62]. HIF-1α is specifically increased in the oxygen-deprived tumor microenvironment, stimulating the activation of genes that are crucial for the development of blood vessel structures [63]. VM is associated with multiple molecular pathways [64]. The function of MMPs is essential, as they break down the extracellular matrix, which helps in the creation of VM channels [65]. Additionally, the process known as EMT plays a crucial role in VM, whereby tumor cells undergo a transformation from their epithelial state to a mesenchymal one, acquiring traits like stem cells [32]. GBM patients with the presence of VM exhibit a negative prognosis and show resistance to standard therapies, such as bevacizumab, which is an antiangiogenic medication [66]. Antiangiogenic therapies specifically focus on the endothelial cells of recently developed blood vessels with the goal of interrupting the blood flow to the tumor [67]. Nevertheless, because of the absence of dependence on endothelial cells, these treatments frequently prove to be unsuccessful against VM. The robustness of VM channels enables tumors to sustain sufficient blood flow and food provision, hence compromising the effectiveness of antiangiogenic medications [68]. To counteract the resistance generated by VM, researchers are investigating innovative therapeutic approaches. Focusing on the routes and elements associated with VM, such as HIF-1α, PI3K/AKT, and MMPs, presents a hopeful strategy [32]. Researchers are also studying substances that can prevent the process of EMT and the acquisition of stem cell-like characteristics in tumor cells [69]. Integrating these tactics with traditional therapies may enhance treatment results for GBM patients [70]. To put it simply, VM is a crucial method of resistance in GBM, allowing tumors to avoid antiangiogenic treatments [26]. Therefore, gaining insight into the fundamental mechanisms and creating specific treatments for VM could greatly improve the efficacy of GBM therapy.

### 2.5. Immune Modulation

GBM is a very aggressive brain tumor that is well known for having a very unfavorable prognosis [3]. Antiangiogenic therapies, namely those that target VEGF, have been utilized to impede the blood flow to tumors, with the goal of limiting their growth [29]. Nevertheless, there is a prevalent occurrence of resistance to these therapies, and one of the primary mechanisms contributing to this resistance is immunological regulation [71]. Antiangiogenic therapies have a substantial effect on the TME, affecting both the innate and adaptive immune responses [72]. These therapies can initially decrease the formation of blood vessels in tumors and restore the natural structure of aberrant blood vessels, which may improve the infiltration of immune cells [73]. Nevertheless, as the body’s resistance to the tumor increases, the immunological environment within the tumor experiences significant alterations that support the tumor’s ability to survive and advance [74].

An essential element of immune regulation in the context of antiangiogenic therapy is the modification of myeloid cell populations [75]. Myeloid-derived suppressor cells (MDSCs) and tumor-associated macrophages (TAMs) are frequently attracted to the TME because of therapy-induced oxygen deficiency [76]. These cells have a dual function: they can assist in the early immune response against tumors, but they frequently develop immunosuppressive characteristics that shield the tumor [77]. MDSCs can prevent the activation and growth of T cells, while TAMs can release immunosuppressive cytokines, including IL-10 and TGF-β, which further weaken the immune response against tumors [78]. In addition, antiangiogenic therapy can create a low-oxygen environment that stabilizes HIFs. HIFs enhance the production of immunosuppressive molecules such as PD-L1 on cancer cells, resulting in T cell exhaustion and compromised immunological monitoring [79]. The presence of hypoxia-induced immunosuppression poses a substantial obstacle to the efficacy of antiangiogenic treatments [80] (Figure 3). Furthermore, the modified blood vessels in resistant tumors frequently hinder the transportation of immune cells and immunotherapeutic drugs, forming a physical obstacle to an efficient immune response [81]. The heightened interstitial fluid pressure and atypical vascular structure diminish the effectiveness of immune cell infiltration and function within the tumor [82]. To summarize, immune regulation is crucial in determining the resistance of GBM to antiangiogenic treatments. The failure of these therapies is attributed to the recruitment and activation of immunosuppressive myeloid cells, the production of immunosuppressive chemicals generated by hypoxia, and the presence of physical barriers that prevent immunological infiltration. Comprehending these pathways is essential for formulating combination methods that can surmount resistance and augment the effectiveness of antiangiogenic therapy in GBM.

## 3. Current Treatment Strategies to Overcome Antiangiogenic Resistance in GBM

GBM is an extremely aggressive kind of brain cancer. Antiangiogenic therapies, which focus on disrupting the blood supply to the tumor, have shown great potential as a therapy method for GBM [83]. Nevertheless, cancers gradually acquire resistance to these therapies. The following is a description of the current approaches employed to overcome this resistance in GBM.

### 3.1. Combination Therapies

Combination therapies have arisen as a hopeful approach to surmount antiangiogenic resistance in GBM [84]. Antiangiogenic therapies, which hinder the development of new blood vessels that nourish the tumor, have demonstrated initial potential but frequently result in resistance and restricted long-term effectiveness [36]. To tackle this issue, researchers are investigating the incorporation of several therapeutic methodologies. An effective method involves combining antiangiogenic drugs with chemotherapy [85]. Temozolomide and other chemotherapeutic medications can augment the effectiveness of antiangiogenic therapy by directly attacking the tumor cells, while the antiangiogenic agents simultaneously obstruct the tumor’s blood supply [86]. This simultaneous assault has the potential to impede or thwart the emergence of resistance. An alternative strategy involves the integration of antiangiogenic treatments with radiation therapy. Tumor cells can have their DNA harmed by radiation, which increases their vulnerability to the impacts of antiangiogenic medications [87]. In addition, radiation therapy can restore the normal structure and function of blood vessels in the tumor, hence enhancing the transportation of antiangiogenic drugs to the tumor location [88]. Immunotherapy is now being investigated in conjunction with antiangiogenic therapies [13]. Combining immune checkpoint inhibitors with antiangiogenic drugs can boost their effectiveness in stimulating the body’s immune response against cancer cells [36]. Antiangiogenic therapy can modify the tumor microenvironment, which may enhance the conditions for immune cells to penetrate and target the tumor [13]. Moreover, employing combination therapies that target several pathways involved in angiogenesis and tumor growth can offer a more comprehensive strategy. By concurrently suppressing many signaling pathways, such as VEGF, PDGF, and integrins, the probability of tumor cells acquiring resistance to treatment is diminished [89]. These combination treatments are being studied in ongoing research and clinical studies with the goal of identifying more potent treatments for GBM. However, medications focus on targeting numerous pathways and mechanisms that contribute to tumor growth and resistance (Table 1). To summarize, combination therapies provide a comprehensive strategy to address resistance to antiangiogenic treatment in GBM, offering potential for improved and long-lasting therapeutic alternatives for patients. Future research is primarily focused on conducting clinical trials to investigate the use of TME-directed therapies in combination with standard therapy [90]. Additionally, there is a need to explore new therapies and GBM models for preclinical investigations.

### 3.2. Personalized Medicine

Utilizing personalized medicine for the treatment of glioblastoma shows potential in overcoming resistance to antiangiogenic treatments [16]. Glioblastoma frequently acquires resistance to therapies such as bevacizumab, an antibody that targets VEGF [84]. Personalized medicine customizes treatments according to the specific genetic and molecular characteristics of a patient’s tumor, resulting in enhanced therapeutic effectiveness and decreased resistance. An important approach involves combining genomic and transcriptomic data to pinpoint certain mutations and pathways that contribute to tumor angiogenesis and resistance mechanisms [96]. Through the process of sequencing the DNA and RNA of the tumor, medical professionals can identify specific mutations and disrupted pathways that can be targeted for treatment [97]. For example, modifications in the PI3K/AKT/mTOR pathway, which are frequently linked to resistance against anti-VEGF therapy, can be effectively addressed by utilizing targeted inhibitors [98]. EGFR inhibitors can be used to treat abnormalities in the EGFR gene [99]. In addition, customized methodologies involve the utilization of liquid biopsies to actively track the response and resistance to treatment in real time [100]. Circulating tumor DNA (ctDNA) and exosomes offer a less intrusive method to monitor the progression of tumors and identify new mutations that may cause resistance to treatment [101]. Furthermore, the potential for success lies in the integration of antiangiogenic therapy with other modalities that are tailored to the specific molecular characteristics of the tumor [16]. Integrating immunotherapy with antiangiogenic drugs can improve treatment response by altering the tumor microenvironment to facilitate the infiltration and activity of immune cells [102,103]. Biomarkers, such as HIFs, can be used to identify patients who are most likely to benefit from combination therapy. Personalized medicine in GBM entails a thorough strategy that utilizes molecular profiling, continuous monitoring, and tailored combination therapy to overcome resistance to antiangiogenic treatment, with the goal of enhancing patient outcomes [104]. GBM can develop bevacizumab resistance through several methods. The tumor may activate alternate pathways to enhance angiogenesis, upregulate growth factors, or become more invasive in hypoxic conditions [51]. This resistance reduces bevacizumab’s efficacy, advancing tumors. To overcome resistance, combination medicines are investigated. Bevacizumab can be used with immune checkpoint inhibitors, metabolic inhibitors, or other targeted medicines that target alternate angiogenesis and tumor development pathways [13]. Developing biomarkers to identify bevacizumab-responsive patients allows for more personalized treatment [105]. Demethylation of the MGMT promoter increases MGMT enzyme production, which repairs TMZ-damaged DNA and allows tumor cells to survive and proliferate [106]. DNA repair pathways, drug absorption and efflux, and epigenetic modifications are other resistance mechanisms. Combining TMZ with MGMT inhibitors to impede DNA repair, targeting alternative DNA repair systems, or using innovative compounds to generate synthetic lethality in tumor cells are ways to circumvent TMZ resistance [107]. Understanding the tumor’s molecular and genetic characteristics can also drive combination therapy or the development of novel drugs that target resistant GBM cells. Combining bevacizumab and TMZ with additional targeted therapy and personalized medicine techniques may help overcome antiangiogenic resistance in GBM, but obstacles remain. The medications included consist of both FDA-approved treatments, and those currently undergoing clinical development are presented in Table 2. This demonstrates the wide range of tactics being studied for personalized medicine in GBM.

### 3.3. Novel Therapeutic Targets

Efforts to address the problem of antiangiogenic resistance in glioblastoma are now mostly centered around the discovery of new therapeutic targets. GBM frequently acquires resistance to antiangiogenic treatments that aim to inhibit the development of new blood vessels that nourish tumor growth [118]. A potential field of study focuses on targeting alternative angiogenesis pathways that circumvent the effects of conventional antiangiogenic medications. For example, medications that target the suppression of non-VEGF pathways, such as those involving angiopoietin-2 or FGF signaling, have demonstrated promise [119]. These pathways play a role in the process of vascular stability and maturation, providing additional targets for VEGF inhibitors. An alternative strategy entails focusing on the tumor microenvironment to augment the effectiveness of antiangiogenic treatments. Researchers aim to disrupt the supporting environment that maintains tumor angiogenesis and growth by modifying immune responses, extracellular matrix components, or metabolic variables inside the GBM microenvironment [120]. Moreover, progress in molecular profiling and genomic sequencing has discovered distinct molecular changes in GBM that could potentially be targeted for therapy [3]. Precision medicine tactics seek to customize treatments by considering the genetic and molecular traits of cancers, which may help overcome inherent resistance mechanisms [27]. Furthermore, there is ongoing research into combination therapies that aim to target various pathways implicated in the angiogenesis and growth of GBM [57]. These combinations may consist of traditional chemotherapeutic drugs, immunotherapies, or targeted drugs that act on both angiogenic and non-angiogenic pathways. The goal is to create synergistic effects and delay or avoid the development of resistance. Ultimately, novel therapeutic targets in GBM seek to challenge the challenges of antiangiogenic resistance by adopting innovative strategies such as targeting alternative pathways, changing the tumor microenvironment, utilizing precision medicine, and implementing combination therapy [121]. Endothelial progenitor cells (EPCs) have the potential to serve as carriers for adenoviral vectors and imaging probes in gene therapy for glioblastoma [122]. These techniques show potential for enhancing results in GBM patients who are unresponsive to traditional antiangiogenic therapies. Bevacizumab, when used in conjunction with temozolomide and radiotherapy, has been established as a conventional therapy [123]. Scientists are investigating the possibility of using different combinations of chemotherapeutic drugs, targeted treatments, or immunotherapies to improve effectiveness and overcome resistance. The objective of combining bevacizumab with immune checkpoint inhibitors, such as nivolumab or pembrolizumab, is to augment the immune response against the tumor and mitigate resistance [124]. The combination of TMZ plus antiangiogenic medicines, such as bevacizumab (a VEGF inhibitor), has been investigated as a means to improve therapeutic effectiveness and prolong the development of resistance [123]. The combination functions by administering chemotherapy to kill the tumor cells while concurrently impeding the development of neovascularization that provides nourishment to the tumor [125]. To overcome resistance, it may be beneficial to target additional pathways, such as MET or EGFR, which are frequently increased in GBM, in combination with bevacizumab [126]. These techniques are now undergoing different phases of research and clinical trials. The primary objective is to enhance the efficacy of current therapies and devise novel strategies to address and overcome resistance in GBM. Table 3 describes synthetic compounds targeting different aspects of the angiogenic process and other pathways critical for GBM cell survival and proliferation.

### 3.4. Immunotherapy

GBM is a highly malignant and refractory kind of brain tumor. Immunotherapy is a promising method to overcome resistance to antiangiogenic treatment in GBM [137]. It involves utilizing the body’s immune system to identify and destroy tumor cells. Checkpoint inhibitors, including PD-1 and CTLA-4 antibodies, have demonstrated promise in counteracting the immune suppression frequently observed in GBM [138]. Targeting tumor-associated myeloid cells through pharmacological means is a promising strategy in the field of developing immunotherapy. Remarkably, myeloid cells exhibit heterogeneity, which includes a specific subgroup of myeloid cells that exhibit angiogenic features in solid tumors [139]. Checkpoint inhibitors can restore the functionality of tired T lymphocytes and enhance their capacity to recognize and eliminate cancer cells by obstructing inhibitory signals [140]. Nevertheless, the efficacy of checkpoint inhibitors in GBM has been constrained by the profoundly immunosuppressive TME [47]. CAR-T cell therapy is a medical procedure that modifies a patient’s T cells to produce receptors that target specific antigens found in GBM [141]. After modifying the T cells, they are reintroduced into the patient’s body. These changed cells can specifically identify and destroy GBM cells. Although CAR-T cell therapy shows potential, it encounters obstacles such as the variability of GBM antigens and the existence of the blood–brain barrier [142]. Cancer vaccines are designed to stimulate a strong immune response against specific antigens found in GBM [143]. Peptide-based vaccinations and dendritic cell vaccines have been investigated for their ability to elicit an immune response against tumors [144]. The EGFRvIII vaccination specifically targets a mutated version of the epidermal growth factor receptor that is frequently present in GBM [145]. Although vaccines have demonstrated some level of success, their efficacy is frequently hindered by the immunosuppressive TME and the tumor’s capacity to avoid immune surveillance [146]. Oncolytic viruses are viruses that have been genetically designed to infect and destroy cancer cells specifically, while also triggering an immune response against the tumor [147]. These viruses can be manipulated to produce immunostimulatory chemicals, which in turn boost the immune response against tumors [148]. Clinical experiments involving oncolytic viruses, specifically the genetically modified herpes simplex virus (HSV), have demonstrated potential in the treatment of GBM by effectively eliminating tumor cells and regulating the immune response [149]. In general, immunotherapy shows potential in overcoming resistance to antiangiogenic treatment in GBM [150]. However, its effectiveness relies on tackling the specific difficulties presented by the TME and enhancing the administration and effectiveness of these therapies. Table 4 provides a concise overview of immunotherapy medications, including information on their molecular mechanism of action, target cells, recommended doses, and current usage in the treatment of GBM.

### 3.5. Nanoparticle-Mediated Treatment Options

Nanoparticle-based therapies offer a hopeful strategy for overcoming resistance to antiangiogenic treatment in GBM [86]. These tactics utilize the distinct characteristics of nanoparticles (NPs) to optimize drug delivery, enhance targeting, and minimize side effects [161]. An important benefit of nanoparticles is their capacity to cross the blood–brain barrier (BBB), a major obstacle in the treatment of GBM [162]. Nanoparticles can be designed to transport diverse therapeutic agents, including chemotherapeutic therapies, small interfering RNA (siRNA), and monoclonal antibodies, directly to the location of the tumor [163]. This precise delivery mechanism aids in retaining a concentrated amount of the therapeutic substance specifically at the tumor location, hence increasing effectiveness and reducing the potential for harmful effects on the entire body [163]. Lipid-based nanoparticles and polymeric nanoparticles have been utilized to provide temozolomide (TMZ), the established chemotherapeutic agent for GBM, therefore enhancing its therapeutic index and surmounting drug resistance [164]. In addition, nanoparticles can be modified by targeting ligands, such as peptides or antibodies, that selectively bind to receptors that are highly expressed on GBM cells or the blood vessels of the tumor [165]. This focused strategy not only enhances the absorption of the therapeutic substances by the cancerous cells but also aids in disturbing the tumor microenvironment, which is vital in overcoming resistance to antiangiogenic treatment. RGD (arginine–glycine–aspartic acid) peptide-functionalized nanoparticles, when used, have demonstrated an increased ability to specifically target integrins that are expressed on GBM cells [166]. This has resulted in improved therapeutic results. Other than delivering medications, nanoparticles can also be engineered to simultaneously transport numerous agents, such as combining antiangiogenic therapies with chemotherapy or immunotherapy [103]. This integrated method can effectively target the tumor from various angles, addressing the complex nature of GBM resistance mechanisms [167]. Therefore, the use of nanoparticles in treatment has great potential to improve the effectiveness of therapies for GBM patients [103]. This is achieved by improved distribution of drugs, enhanced targeting abilities, and overcoming resistance to antiangiogenic treatments. Table 5 provides a concise overview of the current treatment approaches for addressing resistance to antiangiogenic therapy in GBM using nanoparticle-based treatment.

Nanoparticle-based medication delivery devices are being investigated in preclinical phase I trials to address these resistance mechanisms [178]. Nanoparticles possess advantageous characteristics such as their diminutive size and modifiable surface features, which enable them to augment the transportation of drugs to the tumor location, enhance drug durability, and facilitate regulated release [179]. Consequently, the effectiveness of antiangiogenic medicines is heightened. Furthermore, nanoparticles can be designed to circumvent the blood–brain barrier (BBB), which is a major obstacle in GBM treatment, thereby enabling increased medication levels within the tumor microenvironment [103]. Current research has concentrated on utilizing different types of nanoparticle platforms, including as liposomes, polymeric nanoparticles, and inorganic nanoparticles, with the purpose of delivering a combination of antiangiogenic medicines and other treatments, such as chemotherapeutic medications or RNA interference molecules [180,181]. These combinations are designed to simultaneously target numerous resistance pathways, thus decreasing the probability of resistance development. For example, the use of polymeric nanoparticles to contain bevacizumab, an anti-VEGF antibody, and a chemotherapeutic drug has demonstrated potential in decreasing tumor growth and invasion in GBM models that are resistant to antiangiogenic monotherapy [103]. Furthermore, researchers are also studying nanoparticles that are specifically engineered to release antiangiogenic medications in response to the acidic microenvironment of tumors [182]. This has the potential to improve the effectiveness of therapies by ensuring that they are delivered directly to the targeted area. In summary, the use of nanoparticles in preclinical phase I research is a breakthrough in addressing the problem of antiangiogenic resistance in GBM. These methods provide a novel opportunity to enhance the effectiveness of treatments for this difficult kind of cancer and have the potential to be tested in clinical trials soon.

## 4. Limitations and Future Perspectives of Antiangiogenic Resistance in GBM

Antiangiogenic therapy, which primarily focuses on inhibiting the VEGF pathways, has shown great potential as a treatment option for glioblastoma, an extremely aggressive brain tumor [183]. Nevertheless, there are numerous limitations that hinder its effectiveness. Glioblastomas demonstrate both inherent and acquired resistance to antiangiogenic treatment. Intrinsic resistance refers to the inherent inability of tumors to respond to treatment, which is caused by genetic or molecular factors [184]. Tumors gradually gain acquired resistance as they adjust to the antiangiogenic conditions [118]. Mechanisms involve the activation of additional proangiogenic pathways such as FGF and PDGF, enhanced invasiveness, and metabolic reprogramming [32]. The use of antiangiogenic therapy can cause a lack of oxygen inside the TME, which triggers adaptive reactions that support the survival and expansion of the tumor [13]. HIFs are increased in response to low oxygen levels, leading to the activation of other angiogenic factors and promoting invasive and metastatic characteristics [56]. This adaptability hinders the long-term efficacy of antiangiogenic therapies. GBM could use non-vascular techniques to support their growth. This includes taking advantage of existing blood vessels or using alternate methods of forming blood vessels, such as vasculogenic mimicry [185]. These changes allow the tumor to circumvent the impact of antiangiogenic therapies. Antiangiogenic medications can have notable adverse effects, such as high blood pressure, bleeding, and blood clotting disorders [186]. The presence of these negative effects can restrict the length of time and amount of medication used in treatment, so diminishing its overall effectiveness.

To overcome the limits of antiangiogenic therapy in glioblastoma, it is necessary to employ novel methodologies and achieve a more comprehensive understanding of tumor biology. Combination therapies involve the use of antiangiogenic drugs in conjunction with other treatments, such as immunotherapy, chemotherapy, or targeted therapy, to potentially increase the effectiveness of the treatment [29]. For example, the combination of VEGF inhibitors with immune checkpoint inhibitors has the potential to counteract the immunosuppressive effects caused by the tumor microenvironment [187]. Biomarker development is essential to identify dependable biomarkers that may be used to predict and monitor the response to antiangiogenic therapy [188]. Biomarkers can be utilized to customize treatment for individuals, so guaranteeing a more tailored approach and potentially enhancing outcomes [189]. Exploring and focusing on alternate angiogenic routes and resistance mechanisms is crucial. Comprehending the interaction between several proangiogenic factors and their function in the advancement of tumors can provide insights for the creation of therapies that target many variables simultaneously [102]. Advanced drug delivery technologies, such as nanoparticles or convection-enhanced delivery, can improve the transportation of antiangiogenic medications to the tumor site [190]. This leads to higher levels of therapeutic agents in the tumor and reduces the negative effects on the rest of the body. Enhancing preclinical models to accurately replicate human glioblastoma and implementing meticulously planned clinical trials are crucial for effectively transferring laboratory discoveries into successful therapeutic therapies [191]. To summarize, antiangiogenic therapies targeting VEGF or VEGF receptors (VEGFRs) were designed and thought to be an effective tool for controlling the growth of GBM. However, recent results of several clinical trials using humanized monoclonal antibodies against VEGF (bevacizumab), as along with tyrosine kinase inhibitors (TKIs) that target different VEGFRs alone or in combination with other therapeutic agents [192,193,194], demonstrated mixed results, with the majority of reports indicating that gliomas developed resistance to the employed antiangiogenic treatments.. Future research should prioritize investigating combination therapy, developing biomarkers, targeting alternative pathways, enhancing drug delivery systems, and conducting rigorous preclinical and clinical trials to optimize the efficacy of antiangiogenic techniques in managing GBM.

## 5. Conclusions

Antiangiogenic treatments encounter substantial obstacles in the treatment of glioblastoma because resistance mechanisms emerge despite their initial potential [18]. GBM has demonstrated resistance to these therapies via multiple mechanisms [195]. These factors encompass redundant angiogenic signaling, in which numerous angiogenic pathways compensate when one is blocked, and heightened invasion and metastasis, in which GBM cells adjust to antiangiogenic pressure by becoming more invasive [121]. Hypoxia-induced resistance is a significant factor in promoting resistance to treatment which occurs because the lack of oxygen in the TME triggers survival pathways that support resistance [23]. Moreover, the phenomenon of VM, in which tumor cells create structures resembling blood vessels, circumvents conventional antiangiogenic systems. Immunological regulation, which refers to alterations in the immunological milieu that promote tumor development and survival, adds complexity to the effectiveness of antiangiogenic therapy. Presently, the strategies employed to combat these resistance mechanisms revolve around a comprehensive and diverse approach. The integration of antiangiogenic therapies with other treatments, including chemotherapy and radiotherapy, in combination therapies has the potential to enhance efficacy [196]. Personalized medicine, utilizing individual genetic and molecular profiles, seeks to deliver more precise and efficient solutions. Immunotherapy, which utilizes the body’s immune system to fight against cancer, is becoming a highly promising approach, especially when combined with antiangiogenic therapies. In addition, treatment options facilitated by nanoparticles provide novel approaches to enhance medication delivery efficiency and minimize systemic toxicity [197]. However, there are still notable constraints that remain, such as the diverse nature of GBM and the intricate mechanisms of resistance. Future perspectives highlight the necessity for ongoing research to comprehend the fundamental biology of resistance and to formulate more accurate and efficient treatment approaches. Utilizing multi-omics techniques and modern technologies will be crucial in addressing these problems and enhancing outcomes for patients with GBM. To conclude, although there has been some advancement, the struggle against antiangiogenic resistance in GBM continues to be a constantly evolving and ongoing challenge, necessitating a collaborative endeavor from the scientific and medical sectors.

## Figures and Tables

**Figure 1 cancers-16-02975-f001:**
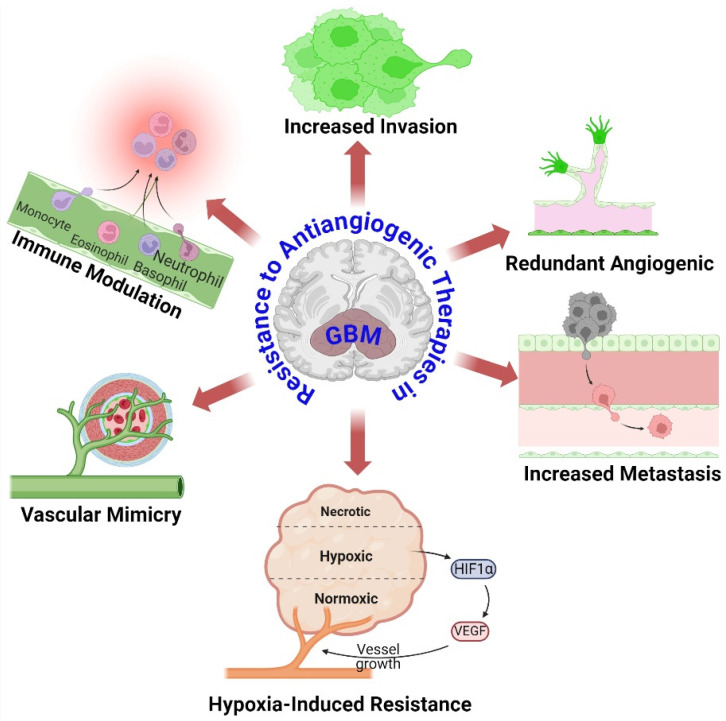
Overview mechanisms of resistance to antiangiogenic therapies in GBM. Despite primary angiogenic signal suppression, GBM tumors activate alternate angiogenic pathways to maintain blood supply and tumor growth. GBM cells become more invasive in response to antiangiogenic therapy, allowing them to colonize distant brain regions and avoid localized therapeutic effects. Antiangiogenic treatment causes tumor microenvironment hypoxia. Hypoxia stabilizes HIFs, which activate genes that promote survival, angiogenesis, and therapeutic resistance. GBM cells transdifferentiate into endothelial-like cells, producing vessel-like structures without angiogenesis and sustaining nutrition supply. Antiangiogenic treatment changes tumor microenvironment immunity. This can recruit immunosuppressive cells and generate immunosuppressive cytokines, allowing tumor cells to avoid immune monitoring and elimination. The figure was drawn by Biorender 16 July 2024.

**Figure 2 cancers-16-02975-f002:**
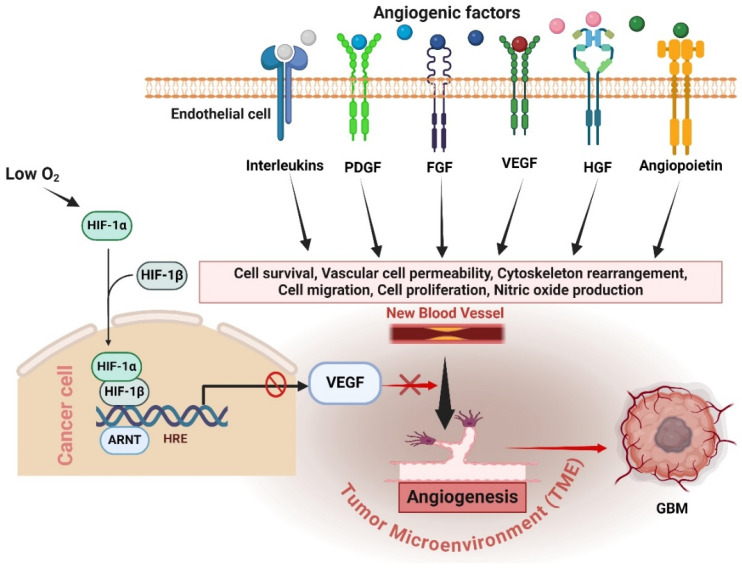
Mechanisms of antiangiogenic resistance in GBM. Anti-VEGF therapies inhibit VEGF, but tumors adapt by activating alternative proangiogenic factors such as FGF, PDGF, HGF, angiopoietins, and interleukins, promoting angiogenesis through VEGF-independent pathways. Hypoxia induced by antiangiogenic therapy triggers HIF-1α, which stimulates various proangiogenic factors, creating complex signaling networks that circumvent VEGF inhibition. Understanding these redundant pathways highlights the challenge of effectively targeting angiogenesis in GBM and underscores the need for combination therapies to manage GBM progression more effectively. The figure was drawn by Biorender 18 July 2024.

**Figure 3 cancers-16-02975-f003:**
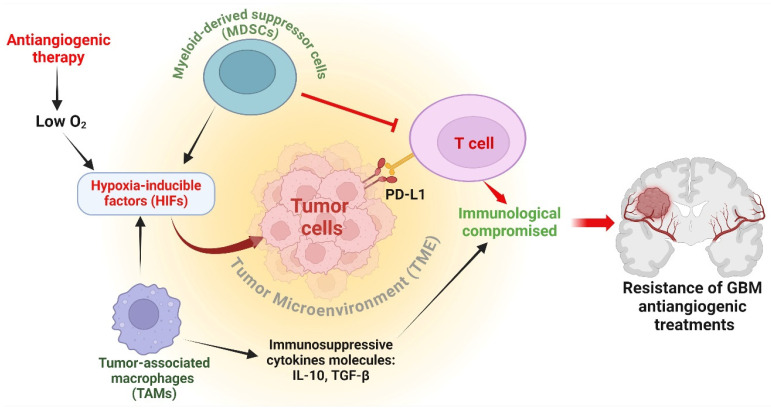
Antiangiogenic immune modulation in GBM. Myeloid-derived suppressor cells (MDSCs) inhibit T cell activation and growth, while tumor-associated macrophages (TAMs) secrete immunosuppressive cytokines IL-10 and TGF-β, impairing immune responses. Antiangiogenic therapy induces hypoxia, stabilizing hypoxia-inducible factors (HIFs) which upregulate PD-L1 on cancer cells, leading to T cell exhaustion. Hypoxia-induced immunosuppression significantly hampers the efficacy of antiangiogenic treatments, highlighting the critical role of immune regulation in glioblastoma (GBM) resistance to these therapies. The figure was drawn by Biorender 18 July 2024.

**Table 1 cancers-16-02975-t001:** Combination medicine therapy targeting GBM.

Drug Name	Molecular Signaling	Target Cells	Doses and Effects	Current Application in GBM	References
Bevacizumab + irinotecan	VEGF inhibition, topoisomerase I inhibition	Endothelial cells, tumor cells	Bevacizumab: 10 mg/kg; irinotecan: 125 mg/m^2^	Used in recurrent GBM; aims to inhibit angiogenesis and tumor growth	[91]
Temozolomide + bevacizumab	DNA alkylation, VEGF inhibition	Tumor cells, endothelial cells	Temozolomide: 150–200 mg/m^2^; bevacizumab: 10 mg/kg	Used in newly diagnosed and recurrent GBM; targets DNA and angiogenesis	[92]
Bevacizumab + lomustine	VEGF inhibition, DNA alkylation	Endothelial cells, tumor cells	Bevacizumab: 10 mg/kg; lomustine: 110 mg/m^2^	Used in recurrent GBM; aims to combine antiangiogenic and cytotoxic effects	[35]
Nivolumab + bevacizumab	PD-1 inhibition, VEGF inhibition	Immune cells, endothelial cells	Nivolumab: 3 mg/kg; bevacizumab: 10 mg/kg	Used in clinical trials for recurrent GBM; aims to enhance immune response	[36]
Bevacizumab + carboplatin	VEGF inhibition, DNA crosslinking	Endothelial cells, tumor cells	Bevacizumab: 10 mg/kg; carboplatin: AUC 5-6	Investigated in recurrent GBM; aims to enhance DNA damage and inhibit angiogenesis	[93]
Pembrolizumab + bevacizumab	PD-1 inhibition, VEGF inhibition	Immune cells, endothelial cells	Pembrolizumab: 200 mg; bevacizumab: 10 mg/kg	Used in clinical trials for recurrent GBM; aims to boost immune system and inhibit angiogenesis	[36]
Bevacizumab + erlotinib	VEGF inhibition, EGFR inhibition	Endothelial cells, tumor cells	Bevacizumab: 10 mg/kg; erlotinib: 150 mg daily	Investigated in recurrent GBM; targets both angiogenesis and EGFR signaling	[94]
Bevacizumab + temsirolimus	VEGF inhibition, mTOR inhibition	Endothelial cells, tumor cells	Bevacizumab: 10 mg/kg; temsirolimus: 25 mg weekly	Used in clinical trials for recurrent GBM; aims to inhibit angiogenesis and mTOR pathway	[14]
Cediranib + lomustine	VEGFR inhibition, DNA alkylation	Endothelial cells, tumor cells	Cediranib: 30 mg daily; lomustine: 110 mg/m^2^	Investigated in recurrent GBM; aims to inhibit angiogenesis and enhance cytotoxicity	[66]
Bevacizumab + ipilimumab	VEGF inhibition, CTLA-4 inhibition	Endothelial cells, immune cells	Bevacizumab: 10 mg/kg; ipilimumab: 3 mg/kg	Used in clinical trials for recurrent GBM; aims to enhance immune response and inhibit angiogenesis	[95]

**Table 2 cancers-16-02975-t002:** Summarizing 10 personalized medicine drugs used in the context of GBM.

Drug Name	Molecular Action	Target Cells	Doses	Current Application in GBM	References
Bevacizumab (Avastin)	VEGF inhibitor, blocks angiogenesis	Endothelial cells	10 mg/kg IV every 2 weeks	Approved for recurrent GBM, reduces edema	[108]
Temozolomide (TMZ)	Alkylating agent, induces DNA damage	Tumor cells	150–200 mg/m^2^/day for 5 days every 28 days	Standard chemotherapy for newly diagnosed GBM	[109]
Everolimus (Afinitor)	mTOR inhibitor, inhibits cell growth and proliferation	Tumor cells	10 mg orally once daily	Under investigation, potential to target mTOR pathway	[110]
Larotrectinib (Vitrakvi)	TRK fusion inhibitor, blocks TRK signaling	Tumor cells with NTRK fusions	100 mg/m^2^ twice daily	Experimental, targeting NTRK fusion-positive GBM	[111]
Enzastaurin (LY317615)	PKC-β inhibitor, induces apoptosis	Tumor cells	500 mg orally once daily	Under investigation, potential anti-tumor activity	[112]
Marizomib (NPI-0052)	Proteasome inhibitor, induces apoptosis	Tumor cells	0.7 mg/m^2^ IV once weekly	Clinical trials, targeting proteasome in GBM	[113]
Abemaciclib (Verzenio)	CDK4/6 inhibitor, inhibits cell cycle progression	Tumor cells	150 mg orally twice daily	Experimental, targeting CDK4/6 pathway in GBM	[114]
Olaparib (Lynparza)	PARP inhibitor, impairs DNA repair	Tumor cells	300 mg orally twice daily	Investigational, for tumors with DNA repair deficiencies	[115]
Nivolumab (Opdivo)	PD-1 inhibitor, boosts immune response	Tumor cells, immune cells	3 mg/kg IV every 2 weeks	Under investigation, potential in immunotherapy	[116]
Toca 511 and Toca FC	Gene therapy, converts prodrug to active chemotherapy	Tumor cells	Toca 511: intratumoral injection; Toca FC: 220 mg/m^2^ orally every 6 weeks	Experimental, gene therapy approach in GBM	[117]

**Table 3 cancers-16-02975-t003:** Synthetic compounds for glioblastoma (GBM) treatment.

Compound Name	Molecular Mechanism	Target Cells	Current Application in GBM	References
Bevacizumab	VEGF inhibitor	Endothelial cells	Approved for recurrent GBM; reduces tumor blood supply	[127]
Temozolomide	DNA methylation/damage	Tumor cells	Standard chemotherapy for GBM; induces cell death	[128]
Cediranib	VEGFR tyrosine kinase inhibitor	Endothelial and tumor cells	Experimental; inhibits angiogenesis	[129]
Sorafenib	Multi-kinase inhibitor (VEGFR, PDGFR, Raf kinases)	Tumor and endothelial cells	Experimental; inhibits cell proliferation and angiogenesis	[130]
Sunitinib	Multi-kinase inhibitor (VEGFR, PDGFR)	Endothelial cells	Experimental; inhibits angiogenesis	[131]
Erlotinib	EGFR tyrosine kinase inhibitor	Tumor cells	Experimental; inhibits tumor cell growth	[132]
Dasatinib	Src family kinase inhibitor	Tumor cells	Experimental; inhibits cell migration and invasion	[133]
Pazopanib	Multi-kinase inhibitor (VEGFR, PDGFR)	Tumor and endothelial cells	Experimental; inhibits angiogenesis and tumor growth	[134]
Regorafenib	Multi-kinase inhibitor (VEGFR, PDGFR, FGFR)	Tumor and endothelial cells	Experimental; inhibits angiogenesis and tumor cell growth	[135]
Axitinib	VEGFR tyrosine kinase inhibitor	Endothelial cells	Experimental; inhibits angiogenesis	[136]

**Table 4 cancers-16-02975-t004:** Immunotherapy medications for the treatment of GBM.

Drug Name	Molecular Action	Target Cells	Doses	Application in GBM	References
Nivolumab (Opdivo)	PD-1 inhibitor	T cells	240 mg every 2 weeks	Investigational; ongoing clinical trials	[151]
Pembrolizumab (Keytruda)	PD-1 inhibitor	T cells	200 mg every 3 weeks	Investigational; ongoing clinical trials	[152]
Ipilimumab (Yervoy)	CTLA-4 inhibitor	T cells	3 mg/kg every 3 weeks for 4 doses	Investigational; combination trials with PD-1 inhibitors	[153]
Avelumab (Bavencio)	PD-L1 inhibitor	Tumor cells, T cells	10 mg/kg every 2 weeks	Investigational; ongoing clinical trials	[154]
Durvalumab (Imfinzi)	PD-L1 inhibitor	Tumor cells, T cells	10 mg/kg every 2 weeks	Investigational; ongoing clinical trials	[155]
Bevacizumab (Avastin)	VEGF inhibitor	Endothelial cells	10 mg/kg every 2 weeks	Approved for recurrent GBM	[156]
Rindopepimut	EGFRvIII-targeted peptide vaccine	Tumor cells	Variable; typically administered intradermally	Phase II/III clinical trials	[157]
DCVax-L	Dendritic cell-based vaccine	Dendritic cells	Personalized; dose varies	Phase III clinical trials	[158]
ONC201	Imipridone; induces TRAIL and DRD2 pathway activation	Tumor cells	625 mg once a week	Phase II clinical trials	[159]
CDX-110 (Rindopepimut)	EGFRvIII-targeted peptide vaccine	Tumor cells	Variable; typically administered intradermally	Phase II/III clinical trials	[160]

**Table 5 cancers-16-02975-t005:** Name of the treatment type of nanoparticle action in mechanism of target in GBM.

Treatment Name	Nanoparticle Type	Mechanism of Action	Target/Focus	References
Bevacizumab-loaded nanoparticles	Lipid-based	Inhibits VEGF, reducing blood vessel formation	VEGF pathway	[168]
Iron oxide nanoparticles	Magnetic	Targets tumor cells via magnetic fields, improving delivery	Hyperthermia, drug delivery	[169]
Curcumin-loaded nanoparticles	Polymeric	Anti-inflammatory and antiangiogenic effects	NF-κB pathway	[170]
Doxorubicin-loaded nanoparticles	Liposome-based	Improves drug accumulation in tumor, reducing angiogenesis	DNA intercalation, inhibiting topoisomerase	[171]
Temozolomide-loaded nanoparticles	Polymeric	Enhances drug delivery and overcomes drug resistance	DNA alkylation, increasing tumor cell death	[172]
Paclitaxel-loaded nanoparticles	Micelle-based	Enhances antiproliferative effects, targeting microtubules	Microtubule stabilization	[173]
siRNA-loaded nanoparticles	Gold nanoparticles	Silences genes involved in angiogenesis and resistance	Gene expression inhibition	[174]
HER2-targeted nanoparticles	Polymer-based	Targets HER2 receptor, enhancing specificity and reducing resistance	HER2 receptor	[175]
Dual drug-loaded nanoparticles	Hybrid (e.g., polymer/lipid)	Combines different mechanisms to enhance therapeutic effects	Multiple targets	[176]
Ceramide-loaded nanoparticles	Lipid-based	Induces apoptosis in resistant cancer cells	Sphingolipid metabolism	[177]

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
