# Peer review of "Recent Treatment Strategies and Molecular Pathways in Resistance Mechanisms of Antiangiogenic Therapies in Glioblastoma"

_cancers, 2024, doi:10.3390/cancers16172975_

Round 1

Reviewer 1 Report

Comments and Suggestions for Authors

All abbreviations should be fully spelled out at first appearances and used after those, even in the abstract.

Bevacizumab and TMZ are included in both Personalized medicine and synthetic compounds, these need more explanations.

Author Response

All abbreviations should be fully spelled out at first appearances and used after those, even in the abstract.

>>Response: Thank you for your valuable feedback. I have carefully reviewed the manuscript and made the necessary changes. All abbreviations have been fully spelled out at their first appearance, including those in the abstract, and are consistently used thereafter with marked by BLUE color.

Bevacizumab and TMZ are included in both Personalized medicine and synthetic compounds, these need more explanations.

>>Response: Thank you for your insightful comment regarding the inclusion of Bevacizumab and Temozolomide (TMZ) under both Personalized Medicine and Synthetic Compounds. We appreciate the opportunity to clarify this aspect of our manuscript. Bevacizumab and TMZ are indeed integral components of glioblastoma therapy, and their categorization under both Personalized Medicine and Synthetic Compounds reflects their dual roles in treatment strategies. By placing these drugs in both categories, we aim to emphasize their significance from both a personalized medicine perspective and as synthetic compounds developed for broad clinical use. Please see page 11 line 384-399, table 2 page 12; and page 13 line 436-451, table 3 page 14.

Reviewer 2 Report

Comments and Suggestions for Authors

Dr. Rahaman and Dr. Ali systematically reviewed the resistance of antiangiogenic therapy in glioblastoma through inherent and adaptive mechanisms, and summarized the current strategies to overcome the resistance. The use of chemotherapy, immunotherapy, targeted therapy, and nanomedicine can sensitize glioblastoma to antiangiogenic therapy, achieving an enhanced therapeutic outcome. This article is beneficial to the field and suitable for publishing. Minor comments are listed below:

1.        It would be helpful to mention how many folds of the resistance increase induced by each pathway, thereby highlighting the critical need to overcome antiangiogenic resistance.

2.        2. Please include the current stage of nanoparticle-mediated treatments, such as preclinical, phase I, etc.

Author Response

Dr. Rahaman and Dr. Ali systematically reviewed the resistance of antiangiogenic therapy in glioblastoma through inherent and adaptive mechanisms, and summarized the current strategies to overcome the resistance. The use of chemotherapy, immunotherapy, targeted therapy, and nanomedicine can sensitize glioblastoma to antiangiogenic therapy, achieving an enhanced therapeutic outcome. This article is beneficial to the field and suitable for publishing. Minor comments are listed below:

>>Response: Thank you for your positive feedback and for recognizing the value of our review article. We appreciate your thoughtful evaluation and are pleased that you find our work beneficial to the field. We have carefully considered your minor comments and have addressed them as follows:

  1. It would be helpful to mention how many folds of the resistance increase induced by each pathway, thereby highlighting the critical need to overcome antiangiogenic resistance.

>>Response: Thank you for your valuable feedback. I agree that quantifying the increase in resistance induced by each pathway is crucial for emphasizing the significance of overcoming antiangiogenic resistance. We have discussed several resistance mechanisms in the manuscript; however, we did not quantify the fold increase in resistance for each pathway. The variability and complexity of these pathways across different studies and models make it challenging to provide a precise fold increase. Additionally, the heterogeneity in experimental conditions and tumor microenvironments may lead to inconsistent data regarding resistance folds. We hope you understand that this complexity limits our ability to include specific quantitative data for each pathway, and we believe the qualitative description sufficiently highlights the critical need to overcome antiangiogenic resistance.

  1. Please include the current stage of nanoparticle-mediated treatments, such as preclinical, phase I, etc.

>>Response: Thank you for your valuable feedback. We have now included a detailed description of the current stages of nanoparticle-mediated treatments for glioblastoma therapy in the revised manuscript. Please see page 17 line 526-549. Additionally, Table 5 provides a concise overview of the current treatment approaches for addressing resistance to antiangiogenic therapy in GBM using nanoparticles-based treatment.

We hope our responses and the corresponding revisions have satisfactorily addressed your comments. Once again, we appreciate your valuable feedback and support for our work.

Reviewer 3 Report

Comments and Suggestions for Authors

The authors review treatment strategies and molecular pathways involved in the resistance to glioblastoma therapies.

The main aspect that caught my attention is the originality of this study with respect to other reviews in the literature. What is new here? A new perspective? New treatments reviewed? The Introduction/Abstract needs to specify clearly the novelty of this review (given that there are so many reviews of glioblastoma treatment approaches already published in the literature).

Author Response

The authors review treatment strategies and molecular pathways involved in the resistance to glioblastoma therapies.

The main aspect that caught my attention is the originality of this study with respect to other reviews in the literature. What is new here? A new perspective? New treatments reviewed? The Introduction/Abstract needs to specify clearly the novelty of this review (given that there are so many reviews of glioblastoma treatment approaches already published in the literature).

>>Response: Thank you for your insightful comments and for recognizing the importance of our review. We appreciate your feedback on the need to clearly specify the novelty of our study in the Introduction and Abstract. In response, we have revised the Introduction and Abstract sections to better highlight the unique aspects of our review.

Specifically, we emphasize that our study provides a comprehensive analysis of the latest advancements in nanoparticle-mediated drug delivery systems for glioblastoma treatment. The current stages of nanoparticle-mediated treatments for glioblastoma therapy in the revised manuscript. Please see page 17 line 526-549.

Unlike previous reviews, our work focuses on the integration of emerging nanoparticle technologies with traditional therapies very recently references (2021 to 2024 mostly), offering a new perspective on overcoming the limitations of current treatment strategies.

Furthermore, we discuss the potential of novel molecular targets that have not been extensively covered in the existing literature with table.

  • Combination medicine therapy targeting GBM (page 9)
  • Personalized Medicine (page 10)
  • Synthetic Compounds for Glioblastoma (GBM) Treatment (page 14)
  • Immunotherapy for Glioblastoma (GBM) Treatment (page 15)

We believe these revisions will clarify the novelty of our review and its contribution to the field. We hope that these changes meet your expectations, and we look forward to your further feedback.

Thank you once again for your valuable input.

Round 2

Reviewer 1 Report

Comments and Suggestions for Authors

I recognized significant improvements in this manuscript.

Reviewer 3 Report

Comments and Suggestions for Authors

The manuscript can be accepted for publication.